# Characterization of Estrogen Receptors in Pancreatic Adenocarcinoma with Tertiary Lymphoid Structures

**DOI:** 10.3390/cancers15030828

**Published:** 2023-01-29

**Authors:** Xuan Zou, Yu Liu, Xuan Lin, Ruijie Wang, Zhengjie Dai, Yusheng Chen, Mingjian Ma, Yesiboli Tasiheng, Yu Yan, Xu Wang, Xianjun Yu, He Cheng, Chen Liu

**Affiliations:** 1Department of Pancreatic Surgery, Fudan University Shanghai Cancer Center, Shanghai 200032, China; 2Department of Oncology, Shanghai Medical College, Fudan University, Shanghai 200032, China; 3Shanghai Pancreatic Cancer Institute, Shanghai 200032, China; 4Pancreatic Cancer Institute, Fudan University, Shanghai 200032, China; 5Cancer Institute, Shanghai Key Laboratory of Radiation Oncology, Fudan University Shanghai Cancer Center, Shanghai 200032, China

**Keywords:** pancreatic adenocarcinoma, estrogen receptor, tertiary lymphoid structure, tumor microenvironment

## Abstract

**Simple Summary:**

The role of estrogen signaling in pancreatic adenocarcinoma (PAAD) was unclear. Here we investigated the expression patterns of three estrogen receptors (ERα, ERβ, and GPER) in 174 PAAD samples via immunohistochemistry staining. Positive expression of all three estrogen receptors was significantly correlated with better clinicopathological characteristics and prognosis, as well as more tertiary lymphoid structure (TLS) presence in PAAD. Upregulated expression of ERα and ERβ in PAAD was also significantly associated with increased CD8^+^ T-cell infiltration in vitro. In-silico analyses also revealed that the expression of estrogen receptors affects multiple pathways relevant to T-cell and B-cell behaviors. In summary, estrogen receptors may remodel the immune microenvironment and regulate the development of TLS in PAAD.

**Abstract:**

The role of estrogen signaling in antitumor immunology remains unknown for non-traditional sex-biased cancer types such as pancreatic adenocarcinoma (PAAD). Tertiary lymphoid structures (TLS) are active zones composed of multiple types of immune cells, whose presence indicates anti-tumor immune responses. In this study, we employed a 12-chemokine signature to characterize potential gene categories associated with TLS development and identified seventeen major gene categories including estrogen receptors (ERs). Immunohistochemistry staining revealed the expression patterns of three ERs (ERα, ERβ, and GPER) in 174 PAAD samples, and their correlation with clinicopathological characteristics, immune cell infiltration levels, and intratumoral TLS presence was analyzed. The results indicated that ERα (+) and ERβ (+) were correlated with high tumor grade, and ERβ (+) and GPER (+) were correlated with lower TNM stage, and both ERα (+) and GPER (+) displayed a beneficial effect on prognosis in this cohort. Interestingly, positive staining of all three ERs was significantly correlated with the presence of intratumoral TLSs and infiltration of more active immune cells into the microenvironment. Moreover, the chemotaxis of CD8^+^T-cells to PAAD cells was significantly increased in vitro with upregulated expression of ERα or ERβ on PAAD cells. To conclude, our study showed a novel correlation between ER expression and TLS development, suggesting that ERs may play a protective role by enhancing anti-tumor immune responses in PAAD.

## 1. Introduction 

Pancreatic adenocarcinoma (PAAD) is one of the most common cancers with poor survival worldwide [1]. Although PAAD is not a traditional gender-biased cancer like breast cancer and prostate cancer, the incidence rate and mortality of male patients are still higher than that of female patients [2]. Gender differences are attributed to enormous factors including genetic, physiological, and environmental factors such as cigarette smoking, obesity, and alcohol intake. At the physiological level, sex hormone pathways undoubtedly have a dominant impact [1].

Estrogens are typical sex hormones that play regulatory roles in multiple physiological processes from reproduction to neuronal development [3]. Emerging evidence indicated that estrogen signaling was extensively involved in regulating cancer cell proliferation, angiogenesis, epithelial-mesenchymal transition, and anti-tumor immunity in multiple tumor types [4,5,6]. Estrogens exert biological effects via two nuclear receptors, estrogen receptor α (ERα) and estrogen receptor β (ERβ) [7], but recent reports have suggested that G-protein coupled ER (GPER) is also involved in the regulation of tumor metabolism and the immune microenvironment [8]. In PAAD, the potential role of ERs has long been debated, and even the expression characteristics of ERs in PAAD are still controversial. The lack of large-sample studies evaluating the expression of different ER isoforms using isoform-specific antibodies (ERα, ERβ, and GPER) likely contributes to the inconsistency of published results concerning ERs expression in PAAD [9]. For example, in a retrospective study of 10 PAAD patients [10], only nuclear ERα expression was detected and found to be expressed in intralobular stromal and islet cells rather than tumor cells in PAAD. Another recently published study identified the broad expression of several ERβ isoforms in 18 PAAD patients [11]. These studies still had rather limited sample sizes and lacked comprehensive analyses of different ERs including both nuclear and membrane ERs.

PAAD is characterized by a complex immune microenvironment, and recent studies have revealed that the successful establishment of adaptive anti-tumor immune responses may be represented by the presence of tertiary lymphoid structures (TLSs) [12,13]. Tertiary lymphoid structures (TLSs) are ectopic lymphoid organs developing in non-lymphoid tissues at sites of chronic inflammation [14]. Intratumoral TLSs are active sites for the generation and activation of innate and adaptive anti-tumor immune responses, and their presence has been shown to be associated with superior prognosis in many cancers [13]. In this study, we first performed bioinformatic analyses to characterize potential regulatory signaling categories for TLS development in PAAD, and the results interestingly revealed a potential link between ER signaling and TLS expression in PAAD. The regulatory roles of estrogen and its receptors in anti-tumor immunity have been gradually revealed [5]. For example, in PAAD, ectopic GPER expression suppressed tumor cell proliferation and normalized the immune microenvironment, indicating the translational value of ERs manipulation for PAAD immunotherapy [15,16,17]. However, few studies have investigated the potential role of ER signaling in the development of tumor-associated TLS structures. 

In this study, we focused on the expression patterns of ERα, ERβ, and GPER, and revealed their correlation with immune status including tumor-infiltrated immune cells and TLS presence in PAAD. For the first time, our study investigated the role of ERs in PAAD prognosis and revealed their potential roles in TLS development. 

## 2. Methods

### 2.1. Data Source 

Transcriptome data of the Cancer Genome Atlas Program (TCGA)-PAAD and the Clinical Proteomic Tumor Analysis Consortium (CPTAC)-PDAC datasets [18] were analyzed in this study. For TCGA-PAAD, fregments per kilobase per million (FPKM) normalized RNA-sequencing (RNA-seq) gene expression data were downloaded and converted to transcripts per million (TPM) format. The Genome Reference Consortium Human Build 38 (GRCh38) assembly was referenced for gene symbol annotation. Gene expression and clinical information matrices of 178 pancreatic adenocarcinoma samples from TCGA-PAAD and 141 from CPTAC-PDAC were analyzed as follows.

### 2.2. Bioinformatic Analysis Methods

TLS score was calculated using a 12-chemokine (CCL2, CCL3, CCL4, CCL5, CCL8, CCL18, CCL19, CCL21, CXCL9, CXCL10, CXCL11, and CXCL13) TLS signature, which was reported as a predictor of TLS expression [19,20], using the single-sample gene set enrichment analysis (ssGSEA) method. For each sample, the TLS expression level was represented by the normalized enrichment score (NES) of ssGSEA result. GSEA analysis was conducted to compare the differentially enriched pathways between two distinct groups, using the classical gene sets from the Kyoto Encyclopedia of Genes and Genomes (KEGG), Gene Oncology (GO), Reactome, and BioCarta databases (https://www.gsea-msigdb.org accessed on 1 June 2022). Spearman correlation analyses were performed to screen the TLS score-correlated genes. The DAVID gene functional classification tool [21] was applied to group functionally related genes from the identified gene list. The abundance of 22 tumor-infiltrated immune cells was inferred from bulk-tissue transcriptome profiles using the CIBERSORTx tool [22].

### 2.3. Patients and Samples

Tumor tissue specimens were collected from PAAD patients receiving upfront surgery between 2012 and 2020 at the Department of Pancreatic Surgery, Fudan University Shanghai Cancer Center. Written informed consent was obtained from each participant. Tissue samples were preserved in formalin-fixed, paraffin-embedded (FFPE) tissue blocks for long-term storage. Hematoxylin and eosin (H&E) staining slides from 348 PAAD patients were first analyzed to investigate the general association between gender, intratumoral TLS presence, and prognosis. The final cohort of 174 samples with complete clinicopathological information and immunohistochemistry staining results was further analyzed to explore the association between ERs expression and clinicopathological features as well as tumor immunity characteristics.

### 2.4. Immunohistochemistry (IHC)

The protein expression level and localization of three ERs and the infiltration level of several immune cells were detected by IHC staining. The primary antibodies used in this study included anti-ERα antibody (1/200, pH 6.0, No. ab79413, Abcam, Waltham, MA, USA), anti-ERβ antibody (1/200, pH 9.0, No. ab288, Abcam, Waltham, MA, USA), anti-GPER antibody (1/200, pH 9.0, No. ab260033, Abcam, Waltham, MA, USA), anti-CD4 antibody (1/200, pH 9.0, No. ab133616, Abcam, Waltham, MA, USA), anti-CD8 antibody (1/100, pH 9.0, No. ab178089, Abcam, Waltham, MA, USA), anti-CD20 antibody (1/100, pH 6.0, No. ab64088, Abcam, Waltham, MA, USA), anti-HLA-DR-antibody (1/200, pH 6.0, No. ab20181, Abcam, Waltham, MA, USA), and anti-FOXP3 antibody (1/100, pH 9.0, No. ab20034, Abcam, Waltham, MA, USA). The accuracy of the anti-ERα, ERβ, and GPER antibodies has been verified in previous studies [23,24,25]. IHC staining was performed following standard procedures of sample dewaxing and hydration, endogenous enzyme removal and antigen repair, blocking, antibody incubation, and DAB staining. Samples with positive staining in more than 10% of tumor cells were regarded as positive for ERs expression. Intratumoral TLS was identified as the regional aggregation of immune cells (mainly T-cells and B-cells) that lacked integrated capsules within tumors on hematoxylin and Eosin (H&E) stained pathology slides, followed by sequential sections stained with T-cell and B-cell markers (CD4, CD8, and CD20) to determine the characteristic cellular compositions and concentric distribution patterns. The relative infiltration level of each immune cell type was calculated as the mean density in more than 3 random sites under a 20-fold microscope magnification.

### 2.5. Immune Cell Chemotaxis Assay

Peripheral blood samples of PAAD patients were diluted in phosphate-buffered saline and added to Ficoll (GE Healthcare, Chicago, IL, USA) for gradient centrifugation to obtain peripheral blood mononuclear cells (PBMCs). CD8^+^T-cells were purified from PBMCs using the Human T-Cell Isolation Kit (STEMCELL Technologies, Vancouver, BC, Canada) according to the manufacturer’s protocol. Human PAAD cell lines BxPC-3 and Capan-2 were cultured respectively in RPMI-1640 medium (Gibco, Waltham, MA, USA) and McCoy’s 5a medium (ATCC) supplemented with 10% fetal calf serum (Sigma, Burlington, NJ, USA) and 10 nM 17 β-estradiol (Sigma, Burlington, NJ, USA) in a humidified atmosphere containing 5% CO2 at 37 °C. ESR1 or ESR2 was overexpressed in human PAAD cell lines via the transfection of pLVX-ESR1(ESR2)-GFP plasmids. The overexpression efficiency was assessed by quantitative reverse transcription-polymerase chain reaction (qRT-PCR) and flow cytometry.

Chemotactic assays were performed to measure the chemotaxis of ERs expressed in PAAD cells on CD8^+^T-cells. CD8^+^T-cells (1 × 10^6^) and human PAAD cells with or without ESR overexpression (1 × 10^6^) were separately seeded in transwell chambers (3 μm, Corning, New York, NY, USA) or at the bottom of chambers. After 24 h of co-culture, the culture supernatant of PAAD cells was collected and the number of CD8^+^T-cells was counted.

### 2.6. Statistical Methods and Software

The correlation between gene expression level and TLS score was measured by spearman correlation analysis. Chi-squared testing was conducted to evaluate the association between ERs expression and intratumoral TLS presence or other clinicopathological features. The survival distribution of samples from two groups was compared by Kaplan-Meier survival analysis. TLS scores and immune cell infiltration levels were compared between two independent groups using Wilcoxon test. “GSVA”, “limma”, and “GSEABase” R packages were used for ssGSEA analysis [26,27]. Statistical analyses were performed on SPSS (version 25.0), Prism (version 7.0), and R (version 4.1.1) software. A two-tailed *p-value* < 0.05 was considered statistically significant.

## 3. Results

### 3.1. Characterization of Regulatory Factors of TLS Development in PAAD

As previously reported, a 12-chemokine signature served as a pan-cancer marker of TLS expression and immunophenotype, and its predictive value had been validated in various types of cancer [19,20,28]. To investigate the potential regulatory signaling of TLS development in PAAD, we first employed the signatures to calculate the TLS scores of samples from two PAAD datasets (TCGA and CPTAC). After that, GSEA analyses were performed to compare groups with high and low TLS scores (the median value as a cutoff) to identify pathways significantly associated with TLS phenotype (Appendix A). A total of 1056 genes whose expression levels were significantly correlated with TLS scores in both datasets were identified and analyzed for gene functional classification (Appendix A). The genes were classified into seventeen groups with distinct biological categories such as cytokine receptors, C-type lectins, and toll-like receptors (Appendix A). In addition, TLS scores were also significantly correlated with the expression of some genes in RAS oncogene family, APOBEC family, and nuclear hormone receptors (including nuclear ER-encoded genes ESR1 and ESR2), etc. (Figure 1A). 

### 3.2. Correlation of Estrogen Receptors with TLS Development

Interestingly, several pathways associated with estrogen signaling (e.g., “extra-nuclear estrogen signaling”, “estrogen-dependent nuclear events downstream of ESR-membrane signaling”, “ESR mediated signaling”, “steroid hormone biosynthesis”) were found to be enriched in the differentially expressed genes between the groups with high and low TLS scores (Appendix A). Particularly, ESR1 and ESR2 (Figure 1B,C) were significantly and positively correlated with TLS scores in both datasets. Meanwhile, female patients displayed significantly higher TLS scores than male patients (Figure 1D,E), suggesting that the presence of TLS may be gender-biased and relevant to the expression of estrogen receptors.

### 3.3. Gender Bias in PAAD Prognosis and TLS Expression

The influence of sex on patient prognosis and TLS expression was first investigated by H&E staining analysis in a large cohort of 348 PAAD patients from Fudan University Shanghai Cancer Center. As shown in Figure 2, the female patients displayed a better overall survival than the male patients (Figure 2A). In this cohort, intratumoral TLS structures were found in 22.1% (77/348) patients, and TLS presence may indicate a more favorable prognosis (Figure 2B). Female patients were found to have a marginally higher proportion of TLS (+) samples (24.4% vs. 20.3%), although no statistically significant difference was found (*p* > 0.05, chi-square test; Figure 2C). Notably, female PAAD patients with positive intratumoral TLS and male patients without TLS represented the best and the worst survival (Figure 2D). The result indicated a marginal trend that the ER signaling played a protective role in PAAD.

### 3.4. Expression Patterns of ERs in PAAD

The expression of three major estrogen receptors, classical nuclear ERs (ERα and ERβ) and transmembrane GPER, were investigated in 174 PAAD samples via IHC analysis. The positive rates of ERβ and GPER expression were 73.0% and 77.0% respectively, while ERα was positively detected in 41.4% of all PAAD samples. Interestingly, although the ERs’ expression in individuals displayed distinct patterns, there were no statistical differences in their positive rates between males and females (Table 1). ERα (Figure 3A) and ERβ (Figure 3B) were expressed in the nucleus and cytoplasm of tumor cells, while GPER (Figure 3C) was mainly located in the cell membrane and cytoplasm. In addition to tumor cells, ERs were also expressed in stromal cells, immune cells, and islets in PAAD tumor tissue.

The association between ERs expression and clinicopathological characteristics was further analyzed. ERα (+) (Figure 3D) and GPER (+) (Figure 3H) patients had better survival than the negative groups, but no significant difference was found for ERβ (Figure 3F). In addition, patients with positive ERs expression displayed more protective clinicopathological features (Table 1). ERα (Figure 3E) and ERβ (Figure 3G) expression were significantly correlated with lower tumor grades. Meanwhile, patients in the early stages had higher rates of positive ERβ (Figure 3G) and GPER (Figure 3I) expression. Together, the results revealed the potential beneficial effects of ERs and ER signaling in PAAD.

### 3.5. Association between Positive Ers Expression and TLS Presence in PAAD

Bioinformatic analyses demonstrated a potential correlation between ER signaling and PAAD-associated TLS development. Here we performed IHC on 174 PAAD samples from the local center to perform further investigation. The concurrence of positive ER expression and intratumoral TLS presence in PAAD could be frequently observed for ERα (Figure 4A), ERβ (Figure 4D), and GPER (Figure 4G). For ERα, 58.3% of ERα (+) samples were TLS-positive simultaneously, significantly higher than that of ERα (−) samples (*p* = 0.0000007, Figure 4B). Compared to the ER (−) group, ERα (+) samples also had significantly higher levels of CD8^+^ T-cell infiltration and higher levels of HLA-DR expression in tumor tissues (Figure 4C). The positive expression of ERβ in tumor cells was also significantly correlated with a higher incidence of TLS presence (*p* = 0.003, Figure 4E) and more CD8^+^T-cells infiltration (Figure 4F) in PAAD. The positive correlation between TLS presence and GPER expression in tumor cells was consistently identified (*p* = 0.001, Figure 4H), although CD8^+^T-cell levels and HLA-DR expression were only marginally higher in the GPER (+) group (Figure 4I). Taken together, ERs expression in tumor cells was statistically associated with an immune-active tumor microenvironment in PAAD.

### 3.6. In Vitro Verification of the Influence of Ers Expression on Immune Cell Chemotaxis

By analyzing the IHC staining from 174 PAAD samples, we found that both the positive expression of ERα and ERβ on PAAD tumor cells were significantly associated with high infiltration of CD8^+^T-cells in the tumor microenvironment. To validate the finding in vitro, we constructed ESR1 or ESR2-overexpressed PAAD cell lines and then used chemotaxis assays to evaluate the influence of ERs expression on CD8^+^T-cell migration. After transfected with ESR1 or ESR2, both the mRNA (Figure 5A) and protein (Figure 5B) expression levels of ESR1 or ESR2 in tumor cells were significantly upregulated. CD8^+^T-cells in the transwell chamber were co-cultured with or without ESR1 or ESR2-overexpressed PAAD cells seeded at the bottom (Figure 5C). After 24 h, the numbers of CD8^+^T-cells migrating to the bottom were counted (Figure 5D). Notably, the chemotaxis of CD8^+^T-cells was significantly increased with upregulated expression of ERα or ERβ on PAAD cells, further demonstrating the significant roles of ERs in anti-tumor immunity in PAAD.

Significant roles of ERs in anti-tumor immunity in PAAD.

### 3.7. In-Silico Analyses of the Influence of Estrogen Receptors on Tumor Immune Microenvironment

The potential immune stimulator function of ERs was further explored in the TCGA-PAAD dataset via immune-cell abundance estimation using CIBERSORTx and functional enrichment analysis using GSEA. CIBERSORTx analysis helped interpret transcriptome data into proportions of 22 tumor-infiltrated immune cells. With this tool, we could verify the findings of IHC experiments from a different perspective through bioinformatic analysis. Spearman correlation analysis was then conducted to assess the potential correlation between ERs-encoded gene expression and immune cell infiltration levels. Consistent with IHC findings, the expression levels of ESR1 and ESR2 were positively and significantly correlated with CD4/CD8^+^T-cell infiltration in PAAD (Figure 6A).

To gain insights into the biological mechanisms of ESR1 and ESR2 in PAAD, samples from the TCGA-PAAD dataset were divided into ESR1(ESR2)-high and low groups based on the median value of ESR1(ESR2) mRNA expression, and GSEA analyses were performed between the two groups. Figure 6B,C show the most significantly enriched pathways in ESR1-high and ESR2-high groups, including the B-cell receptor signaling pathway, mature B-cell differentiation, T-cell migration, positive T-cell selection, T-helper 17 type immune response, etc. Since B-cells and T-cells were the predominant cell types in TLS structures, the results identified significantly up-regulated T/B-cell-related pathways in PAAD with high ERs expression, which further demonstrated the potential stimulatory role of ER signaling in PAAD-associated TLS formation and function.

## 4. Discussion

### 4.1. Prognostic Value of ERs in Non-Traditional Sex-Biased Cancer

Recent studies revealed gender disparities in the incidence and mortality rates of multiple cancers. For most cancers, including PAAD, males have a higher incidence and worse prognosis than females of all ages and races [29]. In addition, there was growing evidence of sex differences in responses to chemotherapy or immunotherapy in certain cancer types [30]. Sex bias in cancers may involve multidimensional mechanisms, including gender-related genetic or epigenetic regulation, and environmental factors. Sex hormones (estrogen in particular) and their receptors were reported to play important roles in cancer biology by affecting cancer stem cell self-renewal, cancer metabolism, and the immune microenvironment [31].

Estrogen, acting through estrogen receptors (nuclear receptors ERα and ERβ and membrane receptors GPER), was extensively deregulated during the development and progression of esophageal cancer [32], gastric cancer [33], and colon cancer [34]. Both elevated estrogen levels and ectopic ERs expression affected the carcinogenesis of certain cancer types, with a variety of outcomes [35,36]. In PAAD, ER signaling was generally reported to be a repressive factor in tumor development, but expression patterns and potential mechanisms of ER signaling are still poorly understood [37,38].

In this study, the influence of gender and TLS expression on patient prognosis was first investigated by H&E staining analysis in a large cohort of 348 PAAD patients from our institute. Through the large sample study, we confirmed the prognostic value of intratumoral TLS in PAAD. In addition, we found that female patients had a slightly better prognosis and a relatively higher proportion of TLS (+) samples than male patients. Together, the result indicated a potential protective role of ER signaling in PAAD.

We then attempted to investigate the expression patterns and assess the prognostic values for all three ERs in PAAD. Positive staining can be detected for ERα, ERβ, and GPER in 41.4%, 73.0%, and 77.0% of PAAD tumor tissues, respectively. In particular, ERα (+) and ERβ (+) were correlated with high tumor grade, ERβ (+) and GPER (+) were correlated with lower TNM stage, and both ERα and GPER showed a beneficial effect in this cohort. These findings revealed that the positive expression of ERs in tumor cells might serve novel beneficial prognostic factors for PAAD.

### 4.2. Estrogen Signaling as a Target to Remodel PAAD Microenvironment

PAAD is featured by a “desmoplasia, inflammation, and immune suppression” microenvironment and recent studies suggested that GPER may harbor the capacity to remodel the microenvironment [15,16,17]. Our study reported that the expression of ERs (including GPER) in PAAD was strongly correlated with the presence of intratumoral TLSs and more infiltration of active immune cells in the tumor microenvironment. In vitro analyses further revealed that the chemotaxis of CD8^+^T-cells was significantly increased under the influence of upregulated ERα or ERβ on PAAD cells, demonstrating the role of ERs in anti-tumor immunity in PAAD.

However, whether ER signaling can serve as a therapeutic target to induce TLSs in PAAD and to remodel the immune microenvironment remains unclear. Recently, the rapid development of molecular dynamics and other computational approaches has greatly facilitated the understanding of dynamic receptor–ligand interactions [39,40]. These tools may become an appropriate complement to understanding the roles of ER signaling in PAAD and identifying potential therapeutics targeting ERs to improve PAAD prognosis.

Nevertheless, the functions of estrogen signaling in regulating immune response were well-understood in both autoimmune diseases and other cancers [41]. Previous evidence demonstrated that ER signaling could contribute to the maturation, activation, or proliferation of a diversity of immune cells (e.g., effector T-cells and B-cells) [42,43,44]. Since intratumoral TLS formation was based on crosstalk among immune cells, further study will focus on the TLS-inducing mechanism of estrogen signaling.

To conclude, our study provided an overview of ERα, ERβ, and GPER expression patterns in PAAD. The expression of all three ERs was correlated with the formation and development of PAAD-associated TLSs, and the manipulation of ER signaling may contribute to the remodeling of the PAAD immune microenvironment.

## Figures and Tables

**Figure 1 cancers-15-00828-f001:**
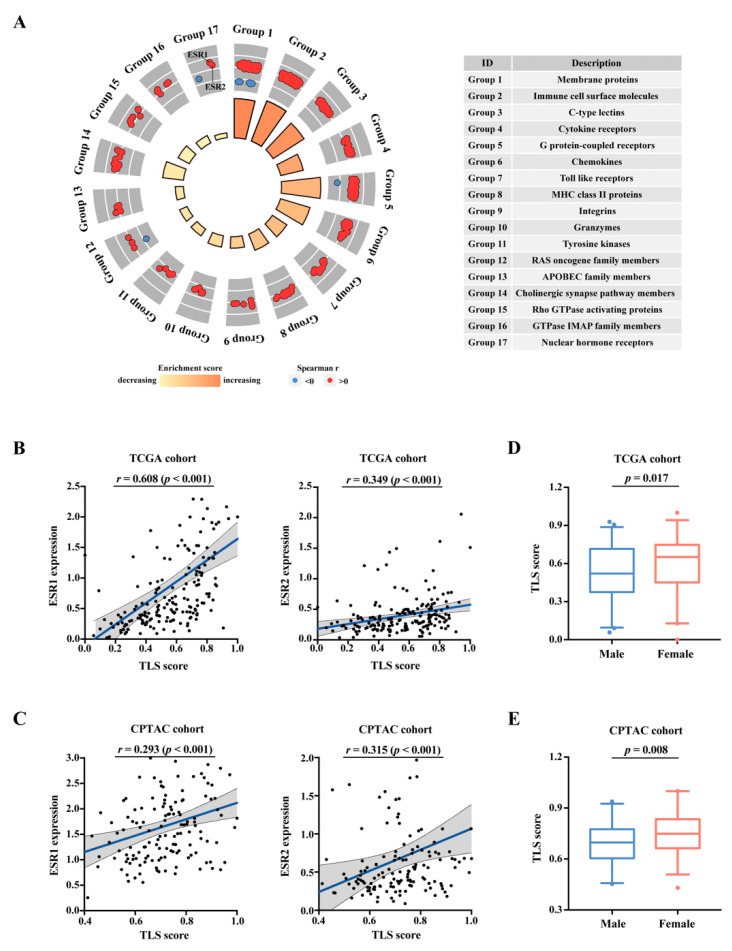
Association between estrogen signaling and TLS scores in two public databases. (**A**) Circular plot showing gene categories significantly correlated with TLS score; the inner circle displays the gene number and enrichment score in each group; the outer circle displays the scatter plots of spearman’s r-values of gene members. (**B**) Scatter plots of the positive correlation between TLS score and ESR1 and ESR2 expression in the TCGA cohort (*n* = 178). (**C**) Scatter plots of the positive correlation between TLS score and ESR1 and ESR2 expression in the CPTAC cohort (*n* = 141). (**D**) Comparison of TLS scores between males and females in the TCGA cohort (*n* = 178). (**E**) Comparison of TLS scores between males and females in the TCGA cohort (*n* = 141).

**Figure 2 cancers-15-00828-f002:**
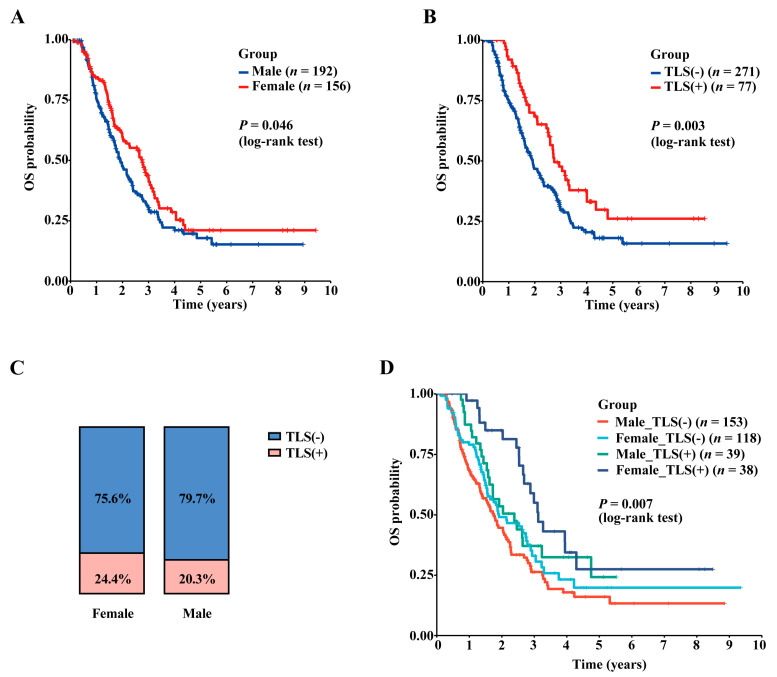
Gender difference of TLS incidences and prognosis in PAAD patients. (**A**) The overall survival curves of male and female PAAD patients. (**B**) The overall survival curves of intratumoral TLS (+) and TLS (−) PAAD patients. (**C**) Proportion distribution of TLS (+) and TLS (−) samples in different gender groups; *p* > 0.05, chi-square test. (**D**) Survival curve analysis based on gender and TLS classification.

**Figure 3 cancers-15-00828-f003:**
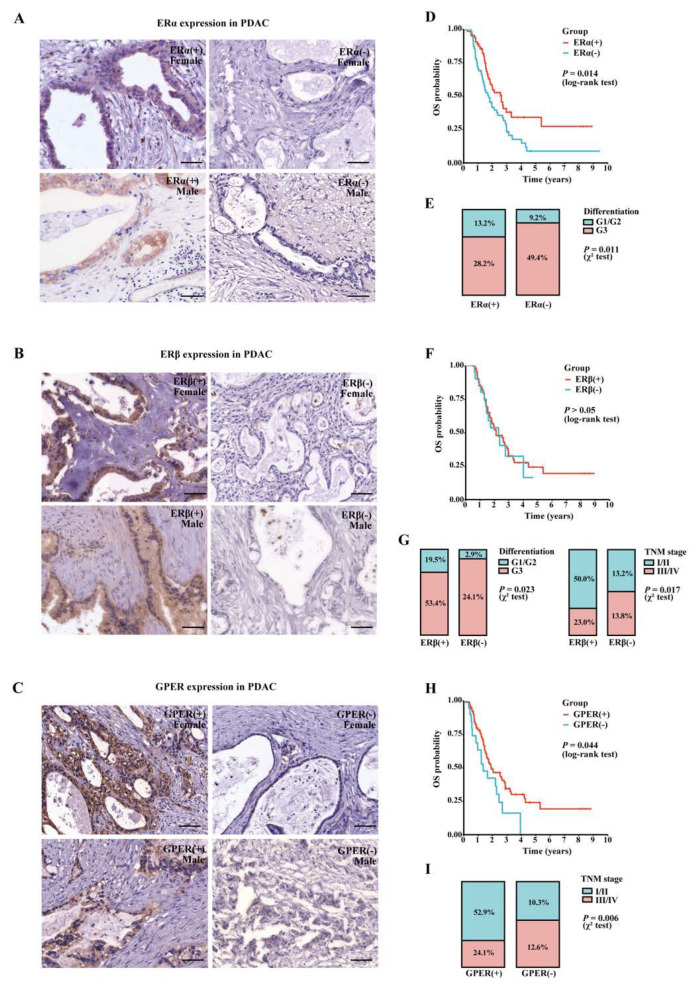
Expression patterns of ERα/β and GPER in PAAD tumor tissues. (**A**) Representative images of ERα staining in PAAD samples from male and female patients; scale bar: 50 µm. (**B**) Representative images of ERβ staining in PAAD samples from male and female patients; scale bar: 50 µm. (**C**) Representative images of GPER staining in PAAD samples from male and female patients; scale bar: 50 µm. (**D**) The overall survival curves of ERα (+) and ERα (−) patients. (**E**) Association between ERα expression and tumor grade in 174 patients. (**F**) The overall survival curves of ERβ (+) and ERβ (−) patients. (**G**) Association between ERβ expression and tumor grade, TNM stage in 174 patients. (**H**) The overall survival curves of GPER (+) and GPER (−) patients. (**I**) Association between GPER expression and tumor grade in 174 patients.

**Figure 4 cancers-15-00828-f004:**
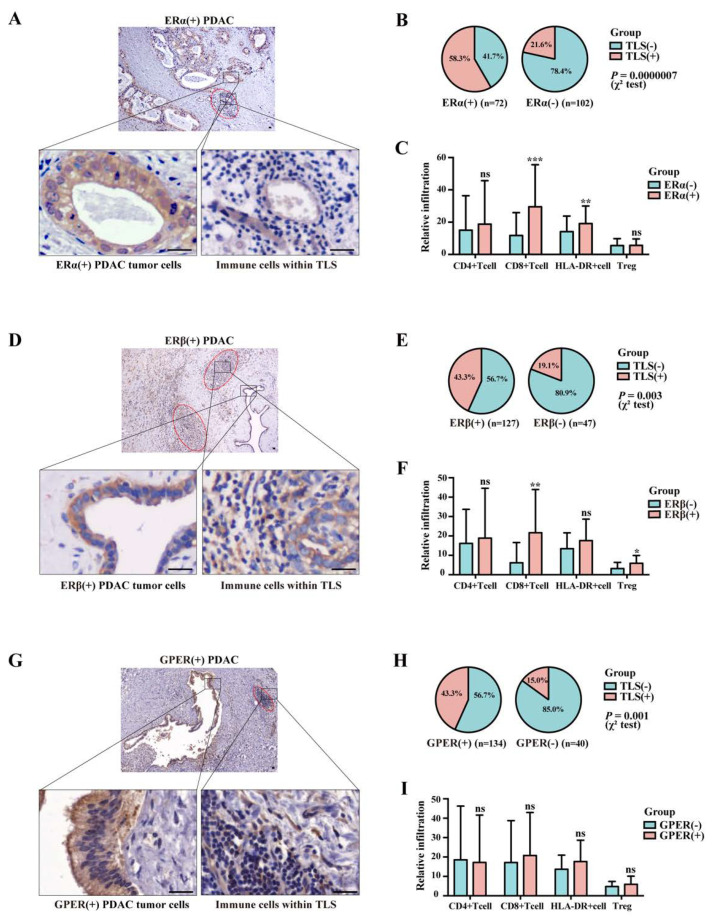
Correlation between ERs expression and tumor-infiltrating immune cells in PAAD. (**A**) Representative IHC images of ERα (+) PAAD tumor tissues with concurrent intratumoral TLS structures; TLS was circled in red line; scale bar: 20 µm. (**B**) Pie plots of the proportion of TLS (+) samples in ERα (+) and ERα (−) groups. (**C**) Histograms of relative infiltration levels of CD4+T-cells, CD8^+^T-cells, HLA-DR+ activated immune cells, and FOXP3+ Tregs in ERα (+) and ERα (−) groups; Treg: regulatory T-cell; ** *p* < 0.01, *** *p* < 0.001, ns: non-significant. (**D**) Representative IHC images of ERβ (+) PAAD tumor tissues with concurrent intratumoral TLS structures; TLS was circled in red line; scale bar: 20 µm. (**E**) Pie plots of the proportion of TLS (+) samples in ERβ (+) and ERβ (−) groups. (**F**) Histograms of relative infiltration levels of CD4+T-cells, CD8^+^T-cells, HLA-DR+ activated immune cells, and FOXP3+ Tregs in ERα (+) and ERα (−) groups; * *p* < 0.05, ** *p* < 0.01, ns: non-significant. (**G**) Representative IHC images of GPER (+) PAAD tumor tissues with concurrent intratumoral TLS structures; TLS was circled in red line; scale bar: 20 µm. (**H**) Pie plots of the proportion of TLS (+) samples in GPER (+) and GPER (−) groups. (**I**) Histograms comparing relative infiltration levels of CD4+T-cells, CD8^+^T-cells, HLA-DR+ activated immune cells and FOXP3+ Tregs in ERα (+) and ERα (−) groups; ns: non-significant.

**Figure 5 cancers-15-00828-f005:**
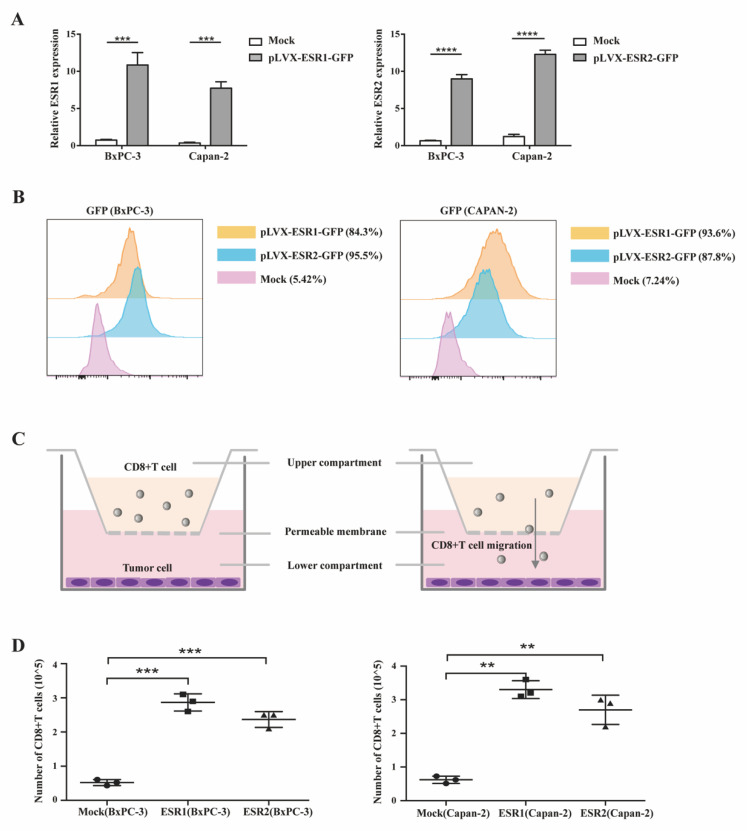
Immune cell chemotaxis assay. (**A**) Expression efficiency of ESR1 and ESR2 in PAAD tumor cells detected by qRT-PCR; *** *p* < 0.001, **** *p* < 0.0001. (**B**) Efficiency of ESR1 or ESR2 (GFP) overexpression in BxPC-3 or Capan-2 cells detected by flow cytometry. (**C**) The schematic diagram of chemotaxis assay. (**D**) The number of CD8^+^T-cells in the supernatant of BxPC-3 or Capan-2 cells in chemotaxis assay; ** *p* < 0.01, *** *p* < 0.001.

**Figure 6 cancers-15-00828-f006:**
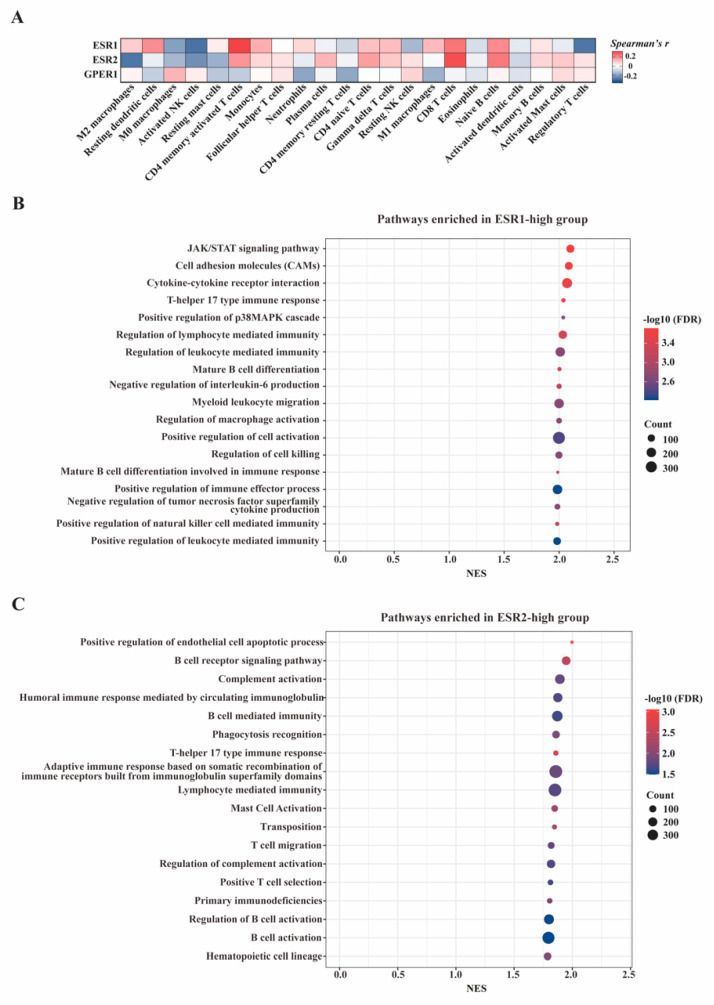
In silico functional exploration of ERs in PAAD. (**A**) Heatmap showing the association between ERs gene expression levels and proportions of 22 immune cell types in tumor microenvironment. (**B**) Diagram of the pathways enriched in the ESR1-high group; the median of gene expression was used as a cutoff value for sample classification. (**C**) Diagram of the pathways enriched in the ESR2-high group.

**Table 1 cancers-15-00828-t001:** Expression of ERs in PAAD and its association with patients‘ clinicopathological features.

Variables	ERα Expression			ERβ Expression			GPER Expression		
	ERα(+)	ERα(−)	Chi-Square Test	ERβ(+)	ERβ(−)	Chi-Square Test	GPER(+)	GPER(−)	Chi-Square Test
**Total**	72(41.4%)	102(58.6%)		127(73.0%)	47(27.0%)		134(77.0%)	40(23.0%)	
**Gender**			0.475			0.821			0.954
Male	37(21.3%)	58(33.3%)		70(40.2%)	25(14.4%)		73(42.0%)	22(12.6%)	
Female	35(20.1)	44(25.3%)		57(32.8%)	22(12.6%)		61(35.1%)	18(10.3%)	
**Age**			0.413			0.628			0.453
≤60	35(20.1%)	56(32.2%)		65(37.4%)	26(14.9%)		68(39.1%)	23(13.2%)	
>60	37(21.3%)	46(26.4%)		62(35.6%)	21(12.1%)		66(37.9%)	17(9.8%)	
**TNM Stage**			0.153			**0.017**			**0.006**
I/II	50(28.7%)	60(34.5%)		87(50.0%)	23(13.2%)		92(52.9%)	18(10.3%)	
III/IV	22(12.6%)	42(24.1%)		40(23.0%)	24(13.8%)		42(24.1%)	22(12.6%)	
**T**			0.470			**0.002**			**0.024**
T1/T2	49(28.2%)	64(36.8%)		91(52.3%)	22(12.6%)		93(53.4%)	20(11.5%)	
T3/T4	23(13.2%)	38(21.8%)		36(20.7%)	25(14.4%)		41(23.6%)	20(11.5%)	
**N**			0.733			0.245			0.826
N0	40(23.0%)	54(31.0%)		72(41.4%)	22(12.6%)		73(42.0%)	21(12.1%)	
N1/N2	32	48(27.6%)		55(31.6%)	25(14.4%)		61(35.1%)	19(10.9%)	
**M**			0.216			0.938			0.090
M0	67(38.5%)	89(51.1%)		114(65.5%)	42(24.1%)		123(70.7%)	33(19.0%)	
M1	5(2.9%)	13(7.5%)		13(7.5%)	5(2.9%)		11(6.3%)	7(4.0%)	
**Grade**			**0.011**			**0.023**			0.200
G1/2	23(13.2%)	16(9.2%)		34(19.5%)	5(2.9%)		33(19.0%)	6(3.4%)	
G3	49(28.2%)	86(49.4%)		93(53.4%)	42(24.1%)		101(58.0%)	34(19.5%)	
**Nerve Invasion**			0.975			0.453			0.439
No	10(5.7%)	14(8.0%)		16(9.2%)	8(4.6%)		17(9.8%)	7(4.0%)	
Yes	62(35.6%)	88(50.6%)		111(63.8%)	39(22.4%)		117(67.2%)	33(19.0%)	
**Vascular Invasion**			0.256			0.221			0.107
No	57(32.8%)	73(42.0%)		98(56.3%)	32(18.4%)		104(59.8%)	26(14.9%)	
Yes	15(8.6%)	29(16.7%)		29(16.7%)	15(8.6%)		30(17.2%)	14(8.0%)	

## Data Availability

All the transcriptome data generated or analyzed during the present study was downloaded from the TCGA (https://portal.gdc.cancer.gov accessed on 10 January 2023) and CPTAC (https://cptac-data-portal.georgetown.edu accessed on 10 January 2023) databases that could be available with open access.

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
