# Peer review of "Characterization of Estrogen Receptors in Pancreatic Adenocarcinoma with Tertiary Lymphoid Structures"

_cancers, 2023, doi:10.3390/cancers15030828_

Round 1

Reviewer 1 Report (Previous Reviewer 1)

The authors appear to have responded appropriately to the reviewers' comments. This reviewer believes that the robust data obtained from future in vivo analyzes of TLS will strengthen the authors' concept.

Reviewer 2 Report (Previous Reviewer 3)

Thank you for considering my comments.

This manuscript is a resubmission of an earlier submission. The following is a list of the peer review reports and author responses from that submission.

Round 1

Reviewer 1 Report

Focusing on the abundance ratio of TLS in cancer (TLS-Score), they are trying to extract gene clusters linked to the formation of TLS. A highly accurate bioinformatic analysis allowed them to find a correlation between TLS and ER family genes. The discovery of a group of genes related to TLS is very interesting, and it is expected that many previously unknown findings will be discovered. However,  the relationship between the ER family gene cluster and pancreatic cancer has been well analyzed previously. In this study, no interventional analysis other than verification of protein expressions with commercially available antibodies has been performed, and only publicly known information has been analyzed. In addition, various results without consensus have been reported regarding the role of ER, including expression levels and outcomes. In this study, which focuses on the ER family, previous reports and their results should be carefully understood to clarify the role of ER genes in pancreatic cancer. To that end, the reviewer believes that it is necessary to carefully verify the accuracy of the antibodies used, as described below, and to demonstrate the concept using experimental studies, such as mouse models.

In the antibody staining performed to verify the bioinformatics analysis from big data, it is necessary to scrutinize whether the antibody reacts correctly. In addition, it is necessary to consider and discuss these results and the data along with the following paper reports.

DOI: https://doi.org/10.18632/oncotarget.26503

DOI: 10.1097/01.mpa.0000226893.09194.ec

DOI: 10.1097/01.mpa.0000226893.09194.ec

DOI10.21037/tgh.2020.02.16

Reviewer 2 Report

In this submission to Cancers, the authors employed a 12-chemokine signature to characterize potential gene categories associated with TLS development and identified seventeen  major categories including estrogen receptors (ERs). The authors carried out IHC staining to reveal the expression patterns of three ERs (ERα, ERβ and GPER) in 174 PAAD samples. The authors' results indicated that ERα 31 (+) and ERβ (+) were correlated with high tumor grade. The authors also found that positive staining of all three ERs was significantly correlated with the presence of intratumoral TLSs and infiltration of more active immune cells in the microenvironment.

I find this manuscript to be of interest to cancer researchers as well as readers of this journal. As such, I am generally in favor of publication with a few minor edits. While the authors focus on experimental studies, there has been much prior research using both molecular dynamics and other computational approaches to understand receptor binding, which have been used to complement these experimental efforts:

Biochemistry 2007, 46, 7, 1743–1758

J. Chem. Theory Comput. 2019, 15, 2807–2815

In particular, these prior studies used large-scale calculations to rationalize binding of receptors and other biochemical systems to give atomistic details into these processes, which should be noted. I am not necessarily asking the authors to carry out such computational studies, but the authors should make note of these as related work in this area. With this minor edit, I would be willing to re-review this manuscript for subsequent publication in Cancers.

Reviewer 3 Report

In a retrospective cohort of patients receiving surgery for PDAC between 2012 to 2020, Zou et al. evaluated: 

1)    the prognostic role of ER expression

2)    the prognostic role of tertiary lymphoid structure

3)    the prognostic effect of gender on prognosis

4)    the influence of ER on tumor microenvironment

Moreover, Authors utilized transcriptome data from two different datasets (TGCA and CPTAC) to calculate a TLS score based on a well characterized 12-chemokine signature  

My observations are the following:

1)    The general aim of the study is not really clearly stated. The sentence more close to an indication of the scope of the study can be found in in line 65 to 68 (“An investigation focusing on ERs expressed in tumor cells and their influence on the establishment of effective antitumor immunology and TLS structure…”). However, in my opinion this is too “general”, and helps in giving the feeling (certainly wrong) of a paper presenting, in part, the results of a “fishing expedition”

2)    The references supporting the statements in line 62-65 (ref. 14 and 15) do not appear appropriate. Indeed,  the first one (Dunnwald et al.) is a prospective cohort study reporting the prognostic of ER in breast cancer, while the second (Sautes-Fridman et al) is a review describing characteristics of TLS and their prognostic-predictive role.  

3)    A clear definition of what the Authors considered to be a Tertiary lymphoid structure is missing. It seems to me that the statement “Intratumoral TLS was identified as the aggregation of immune cells within tumor area” is way too generic. 

4)    The Authors do not indicate a cut-off for the stratification of cases in high and low TLS

5)    At lines 149-152, Authors claim a significant correlation between TLS score and genes of the nuclear hormone receptors. However, from figure 1 and table S2 (wrongly indicated in line 152 as “Table S3”) it is quite evident that such a correlation exists only for ESR1 in the TCGA cohort. In all other cases (ESR2 in TCGA cohort; ESR1 and ESR2 in both cohorts), correlation coefficients range between 0.29 and 0.35. Although not entitled in evaluating statistics, I think that coefficients of this magnitude actually indicate  a weak correlation. Moreover, if I’m not mistaken, P-value associated with Spearman’s r refer to the probability of obtaining such a coefficient by chance, and not to the significance of the correlation between two variables.

6)    Authors report the existence of a gender bias in terms of prognosis and TLS expression. It is not clear to me what this find affects the overall results of the study. In other words, it seems to be an incidental finding. Moreover, as noted by Authors, the observed difference in TLS expression between male and female is not statistically significant. 

As prognosis concerns, the study indicate a better long-term survival for female. However, data are from retrospective sets and as such should be regarded with all the caveats that retrospective studies require.

It follows that their results are not sufficient to “validate the existence of a gender bias in terms of TLS development and prognosis” as stated in lines 185-186.

Moreover, it is not clear what the described gender biases adds to the study, as it was completely ignored in the discussion 

7)    Overall, prognostic role of ER and TLS has been assessed in a univariate fashion. It is evident that several covariates should have been taken into account, and this could be behind the aims of the study. However, I think that for this kind of setting (operable PDAC) it cannot be ignored the effect of stage and adjuvant treatment. Authors, however, give no detail on these aspects. 

8)    At lines 273-275 Authors claim the fact that “expression levels of ESR1 and ESR2 were positively and significantly correlated with CD4/CD8+ T cell infiltration”. However, as reported in figure 5A, this claim is  based, ones again, on correlation coefficients ranging from 0.2 and -0.2.

9)    A cut-off to separate ESR1/ESR2 low and high groups is not provided

10) Based on differences in the pattern of activated pathways between ESR1/ESR2 high vs low group, Authors claim that their results indicated that “ER signaling broadly regulated the behaviors of immune cells and the formation of TLS”. However, no mechanistic prove of this claim as been provided: “association is not causation”